# A Complex Environmental Water-Level Detection Method Based on Improved YOLOv5m

**DOI:** 10.3390/s24165235

**Published:** 2024-08-13

**Authors:** Jiadong Li, Chunya Tong, Hongxing Yuan, Wennan Huang

**Affiliations:** 1School of Cyber Science and Engineering, Ningbo University of Technology, Ningbo 315211, China; sljd37383@gmail.com (J.L.); yuanhx@mail.ustc.edu.cn (H.Y.); 2School of Electronic and Information Engineering, Ningbo University of Technology, Ningbo 315211, China; mohn265358@gmail.com

**Keywords:** water-level recognition, water-level ruler, improved YOLOv5m, skew correction, complex environments

## Abstract

The existing methods for water-level recognition often suffer from inaccurate readings in complex environments, which limits their practicality and reliability. In this paper, we propose a novel approach that combines an improved version of the YOLOv5m model with contextual knowledge for water-level identification. We employ the adaptive threshold Canny operator and Hough transform for skew detection and correction of water-level images. The improved YOLOv5m model is employed to extract the water-level gauge from the input image, followed by refinement of the segmentation results using contextual priors. Additionally, we utilize a linear regression model to predict the water-level value based on the pixel height of the water-level gauge. Extensive experiments conducted in real-world environments encompassing daytime, nighttime, occlusion, and lighting variations demonstrate that our proposed method achieves an average error of less than 2 cm.

## 1. Introduction

In recent years, there has been a rise in the frequency of floods and droughts globally, leading to significant damage to cities due to severe weather events. Particularly during typhoons and heavy rainstorms, the sudden increase in water levels has resulted in substantial losses for urban areas. For instance, the 2022 Typhoon “Fireworks” caused river and seawater to overflow in various parts of Ningbo City, Zhejiang Province, China, leading to water accumulation in low-lying regions and causing economic losses. Consequently, there is an urgent requirement for a reliable and accurate water-level monitoring system.

The evolution of water-level monitoring techniques [1,2,3] has progressed through three stages: manual observation, sensor-based monitoring, and computer vision-based [4] measurement. Manual observation readings, which involve human visual assessment, have been the traditional method for obtaining water-level data. However, this approach is prone to inaccuracies, safety risks, and limitations in adverse weather conditions. Sensor monitoring encompasses both contact and non-contact methods. Contact sensors, such as float and pressure group types, are susceptible to errors due to water impurities, leading to malfunctions and high maintenance costs. Non-contact sensor technologies, including laser, radar, and ultrasonic methods, offer alternatives but are costly and prone to equipment damage, hindering their widespread practical application.

Compared to the first two monitoring methods, computer vision-based water-level measurement has the advantages of low cost, real-time capability, and high accuracy. In the early stages, many researchers used traditional image processing methods [5] for water level detection, including edge detection [6,7,8,9] and machine learning [10]. For example, Sun et al. [6] utilized edge detection algorithms to preprocess captured images, locating the water gauge through edge detection and correcting it via affine transformation. They applied two strategies to process key points and used edge features to detect the water surface position, ultimately calculating the water surface height based on the processed results and edge features. Zhang et al. [9] proposed a water level detection method based on the Maximum Mean Discrepancy (MMD) of grayscale and edge features, effectively overcoming the limitations of traditional single-feature detection methods under complex lighting conditions such as low light, water splashes, and shadow projections. However, in complex environments or when the water gauge is obstructed, this method may produce significant errors. With the rapid development of deep learning, methods based on CNN [11,12,13,14], LSTM [14,15,16,17], and Transformer [18] have gradually been applied to the field of water level detection. Since water level detection involves object recognition [19], some researchers have introduced YOLO [20,21] into their detection schemes. For instance, Qiao et al. [22] proposed a method based on YOLOv5s that uses image processing techniques to detect the position of the water surface line and subsequently accurately calculate the actual water level. Later, other researchers adopted combined methods [23]. Chen et al. [24] proposed a CNN-based hybrid approach for water level recognition, which can obtain accurate water level values in complex and dynamic scenes. However, this method is challenging to apply to embedded AI devices. Xie et al. [25] introduced an innovative water level measurement method that combines an improved SegFormer-UNet with a virtual water gauge, addressing the limitations of previous computer vision methods that relied on water gauge information.

This paper proposes an automatic water-level detection method suitable for complex environments to address the existing research issues. Firstly, use the object detection algorithm to extract the region of interest of the water-level gauge, to reduce the interference of the surrounding environment on the algorithm. Then, the threshold Canny operator and Hough line transformation are used to automatically obtain the tilt angle of the water-level gauge in the image, and the image is corrected through affine transformation. Next, based on the YOLOv5m model, we will improve the attention mechanism and propose a water-level gauge segmentation algorithm based on the improved YOLOv5m, enabling the model to more accurately identify and segment water-level gauges. In addition, a context-based water-level detection auxiliary strategy is proposed to handle extreme situations, such as occlusion and sudden changes in lighting, while compensating for the performance shortcomings of segmentation algorithms, reducing errors, and improving algorithm stability. Finally, the obtained water-level gauge height information will be transformed into water-level values using a linear regression model.

## 2. Related Work

### 2.1. Dataset

The source of experimental data is images collected from a certain location in Ningbo, with a collection time of 10 days and every 5 min. A total of 2556 images were collected. Among them, the first 1776 images were used as training data, and the last 780 images were used as testing data. Collecting images includes various weather conditions, such as sunny, cloudy, rainy, and typhoon days, covering the period from day to night, and covering extreme situations such as rapid changes in water levels and mud blockage. The dataset includes various conditions with the following distribution: 17% for Sunny, 16% for Cloudy, 1.4% for Rainy, 37.9% for Night, 26% for Soiling, and 1.7% for Special circumstances. The data samples for various conditions are shown in Figure 1. Due to significant differences in water-level gauge recognition and segmentation between day and night environments, to better handle these differences, the LabelImg annotation tool is used to annotate the images. Divide the tags into “draft-d” (representing daytime images) and “draft-n” (representing nighttime images), and generate the corresponding XML files. Then, convert it into the format of the VOC dataset. The prepared VOC dataset is divided into training and testing sets, with the training set accounting for 80% and the testing set accounting for 20%. In other words, for every four images used for training, the next image is used for validation.

### 2.2. Water-Level Detection

Figure 2 shows the overall framework of water-level detection, which includes four main processes: image correction, water-level gauge segmentation, context-assisted adjustment strategy, and water-level value calculation.

#### 2.2.1. Image Correction

In an ideal scenario, the water-level gauge in the captured image should be strictly vertically aligned with the horizontal plane. However, due to the camera not being placed strictly vertically or being swayed by factors such as wind, there may be a non-90-degree angle between the water-level gauge and the horizontal plane in the image. When the camera is tilted, image perspective distortion may occur. However, since this study uses a Hikvision standard wide-angle lens, the perspective distortion in images captured within the central area of the camera, approximately 50–60 degrees, can be ignored. To avoid errors caused by the tilt of the water-level gauge, it is necessary to correct the image. Jiang et al. [7] proposed a remote system for water-level monitoring using smartphones, which combines image connectivity and image processing to locate the position of the water-level gauge. Based on the minimum area in the binary image, the water-level gauge tilt is corrected. However, this method may fail in complex background environments. The correction method in this article can be effectively applied in complex scene environments. The specific steps are as follows:(1)Using object detection methods to extract regions of interest and extracting the area where the water-level gauge is located from the image to reduce external environmental interference.(2)Perform grayscale transformation on the extracted region of interest image and use the Canny operator to automatically determine the threshold to convert it into a binary image.(3)Use the Hough line transformation function to process binary images, obtain a large number of tilt angle values, and then calculate the average of these tilt angle *θ* values as the tilt angle of the water-level gauge in the image.(4)Finally, in Figure 3, the coordinates of the top-left corner of the bounding box are (*x*_0_, *y*_0_), with the width and height of the bounding box being *w*_0_ and *h*_0_, respectively. From these values, the coordinates of the rotation center can be determined as (*x*_0_ + *w*_0_/2, *y*_0_ + *h*_0_/2). Then, by applying OpenCV’s affine transformation function, the original image is rotated around this rotation center point by the calculated tilt angle *θ*. This rotation aligns the water gauge vertically with the horizontal plane, resulting in a corrected image of the water gauge. The principles of this transformation are illustrated in Formulas (1) and (2) below.
(1)$$M=\mathrm{getRotationMatrix}2D((x_{0}+w_{0}/2,y_{0}+h_{0}/2),\theta )$$
(2)$${\mathrm{img}}_{\mathrm{on}}=\mathrm{warpAffine}({\mathrm{img}}_{\mathrm{in}},M,(w,h))$$
where *getRotationMatrix*2*D* (·) is the function for obtaining the rotation matrix, *M* is the rotation matrix, *warpAffine* (·) is the affine transformation function, and *img_in_* and *img_on_* are the original image and corrected image, respectively.

**Figure 3 sensors-24-05235-f003:**
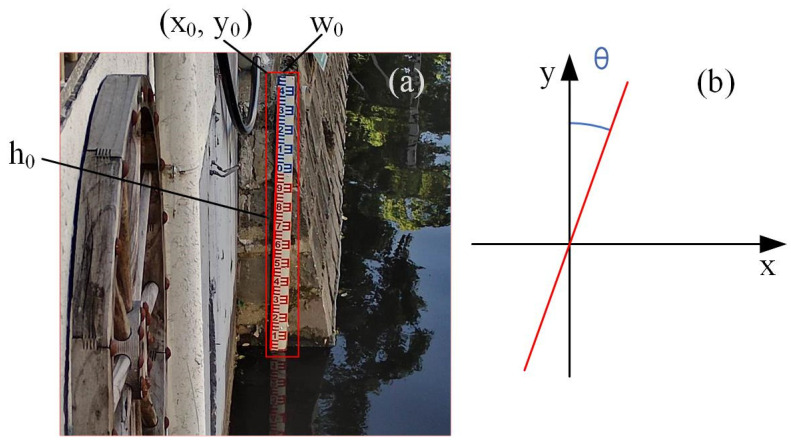
The effect of tilt detection of water-level gauge. (**a**) The straight line was detected by the Hough transform and the top-left corner of the bounding box. (**b**) Schematic diagram of detected tilt angle *θ*.

#### 2.2.2. Water-Level Gauge Segmentation

At present, research on image-based water-level measurement methods mainly uses scale characters containing water-level gauge images to obtain water-level values. For example, Qiao et al. [22] proposed a method based on YOLOv5s that involves extracting the water gauge area and all character regions at various scales from raw video images. Using image processing techniques, they identify the position of the water surface line to calculate the actual water-level value. In most scenarios, the error of this method is less than 0.01 m. However, in transparent water quality and when the water-level gauge has dirt-blocking key characters, it can affect the recognition of scale characters, leading to significant errors in the predicted water-level values. To address this issue, this article adopts another method of obtaining water-level values. The principle of this method is to use the improved YOLOv5m algorithm to segment the water-level gauge in the image, to obtain the height of the water-level gauge. Next, the height of the water-level gauge is converted into a water-level value through linear regression. This method can effectively avoid the influence of factors such as water-level gauge type, external environment, lighting, and surface dirt on the water-level gauge.

#### 2.2.3. Content Assisted Adjustment Strategy

Under normal conditions, the rise and fall of the water level within a specified time frame have a certain threshold. When the external environment’s minimum illuminance drops below 0.01 Lux, it indicates nighttime, and the camera’s infrared function automatically activates, resulting in a black-and-white image. When the water gauge is obscured at night, the obscured part blends with the background, causing the improved YOLOv5m algorithm to only segment the visible, unobstructed part of the water gauge. Consequently, the portion behind the obstruction cannot be detected, leading to an underestimated height of the segmented water gauge and an overestimated calculated water level. The situation is shown in Figure 4.

This paper proposes a context-aware strategy that compares the height difference of the water gauge between consecutive frames with a set threshold to determine if the image is in the described situation. If such a condition is detected, corresponding measures will be taken to adjust the detected water-level value. The basic principles of these measures are as follows:(3)$$H_{\varepsilon }=H_{\mathrm{day},\infty }-H_{\mathrm{night},0}$$
(4)$$\left\{\begin{array}{c} H_{\mathrm{night},t}=H_{\mathrm{night},t}+H_{\varepsilon }\begin{array}{cc} ; & H_{\varepsilon }>H_{o} \end{array}\\ H_{\mathrm{night},t}=H_{\mathrm{night},t}\begin{array}{cc} ; & H_{\varepsilon }\leq H_{o} \end{array} \end{array}\right.$$
where *H_ε_* is the height of the obstruction in extreme situations, *H*_*day*,∞_ is the water-level gauge height of the last frame of the image during the day, *H*_*night*,0_ is the water-level gauge height of the first frame of the image at night, *H_o_* is the set threshold, and *t* represents the number of frames.

#### 2.2.4. Water-Level Calculation

The height of the water-level gauge in the image will change linearly with the change in water level, so a linear regression model is used to predict the water-level value. The steps for predicting water-level values are as follows:(1)Create a training set that includes the actual water-level gauge heights and corresponding water-level values in the images. By normalizing the data, it is limited to the range of from 0 to 1 to eliminate the potential adverse effects caused by singular sample data.(2)Establish a linear regression model.(3)Train and test the model to obtain the optimal model and obtain the weight as w and the bias as b of the model.

## 3. Algorithm

### 3.1. Water-Level Gauge Correction Algorithm

#### 3.1.1. Automatic Determination of Threshold Canny Algorithm

This article uses the Canny operator Sobel operator Laplacian operator and other algorithms to perform image binarization on water-level gauge images, among which the Canny operator is a multi-stage optimization operator with denoising, enhancement, and localization functions. Therefore, this paper improves upon the Canny operator. The improvements are as follows:(1)Gaussian filtering is applied to grayscale images to reduce the impact of noise.(2)Unlike the unimproved Canny operator, the improved high and low thresholds are calculated based on the median of the grayscale image. As shown in Formulas (5) and (6).
(5)low=(1−σ)×median(f)
(6)high=(1+σ)×median(f)
where *f* is the grayscale image, *media* is the median function, *low* is the low threshold, *high* is the high threshold, and σ is set to 0.33. As shown in Figure 5, in both normal and special scenes, when σ is less than 0.33, the algorithm will find it difficult to detect the edge features of the water-level gauge σ. When it is greater than 0.33, the edge features detected by the algorithm will have a lot of noise, which will have a significant impact on Hough line detection.
(3)Perform Gaussian filtering again on the binarized image obtained from the Canny operator to further reduce the impact of noise.

**Figure 5 sensors-24-05235-f005:**
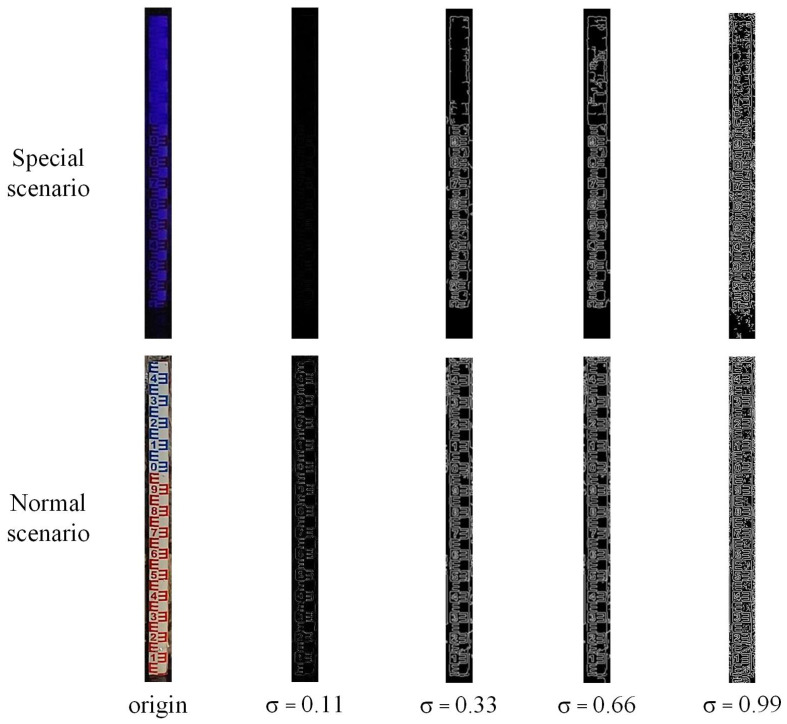
Binary effects of autoCanny with different σ values in normal and special scenarios.

#### 3.1.2. Hough Line Transformation Function

Hough line transformation is a method used to detect lines in image space. This article employs two types of Hough line transformation functions: HoughLinesP and HoughLines. The process involves mapping the input binary image to the Hough space, setting a threshold based on local maxima, filtering out interfering lines, and obtaining the target line. The principle of line detection is illustrated in Formula (7). This method boasts strong anti-interference capabilities, can tolerate gaps in feature boundary descriptions, and is resilient to image noise.
(7)Rho=x∗cos(θ)+y∗sin(θ)

The lines detected by HoughLinesP are output in vector representation sets, and each line is represented by a vector of four elements (*x*1, *y*1, *x*2, *y*2), where (*x*1, *y*1) represents the starting point of the line segment and (*x*2, *y*2) represents the endpoint of the line segment. The lines detected by HoughLines are also output in vector representation sets, with each line represented by a vector with two elements (*Rho*, *θ*). Where *ρ* represents the length of the line from the origin (0, 0), *θ* represents the angle of the line in radians.

### 3.2. Improved YOLOv5m

#### 3.2.1. Overall

The YOLOv5m in this article consists of four parts: Input, Backbone, Neck, and Head. The overall structure diagram of the improved YOLOv5m mentioned in this article is shown in Figure 6.

#### 3.2.2. Input

In the input section, the improved YOLOv5 uses Mosaic data augmentation, where four images are randomly selected and then concatenated based on reference points through random scaling, cropping, and arrangement. Mosaic is beneficial for improving the detection of small targets, as small targets are generally unevenly distributed in the image in the dataset, resulting in insufficient learning of small targets in conventional training. After using Mosaic data augmentation, even if there are no small targets, the original target size becomes closer to the size of the small target by shrinking, which is beneficial for model object detection. Additionally, YOLOv5 uses parameter-based adaptive anchor box computation, which automatically learns and generates optimal anchors based on the training data.

#### 3.2.3. Backbone

In the Backbone section, the Conv module is composed of Conv2d BatchNorm2d and SiLU are used to perform convolution, normalization, and other operations on the input feature map, and, finally, output it through the SiLU activation function. SPPF specifies a convolution and sequentially passes through three multi-scale max-pooling layers, which enhances the receptive field at a higher speed than SPP, helping to solve the alignment problem between anchor points and feature layers. The focus of the Backbone section is on the Bottleneck module, which first reduces and then expands the number of channels. The specific operation is to first perform 1 × 1 convolution to reduce the number of channels by half, and then double the number of channels through 3 × 3 convolution to obtain image features. By using add instead of Concat for feature fusion, the fused features remain unchanged; two 1 × 1 convolutions are used to reduce and increase the feature dimension, respectively, with the main purpose of reducing the number of parameters, thereby reducing computational complexity and enabling more effective and intuitive data training and feature extraction after dimensionality reduction.

#### 3.2.4. Neck

In the Neck section, YOLOv5 adopts an FPN + PAN structure. FPN is top-down, which transfers and fuses high-level feature information through upsampling to improve the detection performance of small targets. Meanwhile, PAN adds a bottom-up connection path based on FPN, which can better transmit the location information from the bottom layer to the top layer. The combination of FPN and PAN conveys strong semantic features from top to bottom through the FPN layer. The PAN structure conveys strong localization features from bottom to top, thereby aggregating features from different detection layers from different backbone layers, playing a role in image segmentation. Moreover, this structure can further improve the detection performance of dense objects and enhance the network’s ability to fuse features of objects of different scales.

#### 3.2.5. Head

In the Head section, this article uses GIOU (Generalized Intersection over Union) as the loss function for bounding boxes and Classification. In the post-processing of object detection, non-maximum suppression (NMS) is used to filter the object boxes, thereby enhancing the detection ability for multiple objects and occluded objects.

#### 3.2.6. Attention Mechanism

In the Backbone section of YOLOv5m, this article has made improvements to the attention mechanism [26,27,28] by replacing the C3 module with the Bottleneck module. As shown in the structure of Bottleneck and C3 in Figure 6, it can be seen that the C3 module has three more standard convolution modules compared to the Bottleneck module, which leads to the overfitting of the model in detecting the water-level gauge. Based on this situation, we reduce the number of parameters and the computational complexity by using the Bottleneck module separately, to prevent overfitting of the model.

### 3.3. Linear Regression

Linear regression, as a statistical method for discovering or estimating possible linear dependencies between two or more variables through data, is a regression analysis that models the relationship between one or more independent and dependent variables using a least squares function called a linear regression equation. This function is a linear combination of one or more model parameters called regression coefficients. A situation with only one independent variable is called simple regression, and a situation with more than one independent variable is called multiple regression.

Because the height of the water-level gauge decreases with the rise in the water level, they exhibit a linear relationship with each other, and it only has a simple regression of one independent variable. The linear relationship is shown in Formula (8) as follows:(8)y=wx+b
where *x* is the height of the water-level gauge in the image and y is the water-level value.

Linear regression models often use least squares approximation for fitting. By estimating the parameters in the model using the training sample set, the model can best fit the training sample in the sense of minimum square error. The mathematical formula is shown in Formula (9).
(9)$$\underset{w,b}{\min}\;E=\frac{1}{N}\sum_{i=1}^{N}{\left(f(x_{i})-y_{i}\right)}^{2}$$

## 4. Experiment and Analysis

### 4.1. Experimental Environment

The development of the deep learning model in this article includes both software and hardware development platforms. In terms of software platform, this article uses Pycharm2021 from ANACONDA3 software for relevant experiments, and the model is constructed using the Pytorch3.8 deep learning framework developed by Facebook Artificial Intelligence Research Institute. In terms of hardware platform, a 64-bit desktop computer using the Ubuntu 18.04.6 operating system, with the processor being an Inter (R) Core (TM) i7-7700KCPU@4.20GHz, and a graphics card with a memory size of 16 GB and model number INVIDA GeForce GTX1080.

### 4.2. Water-Level Gauge Correction Experiment

This article uses four edge detection operators and two Hough line transformation functions to correct the water-level gauge. Then, experimental comparisons and analyses were conducted on the combination of four binarization algorithms and the Hough line transformation function. In Figure 7a,b, the horizontal axis represents the number of images, while the vertical axis represents the detected tilt angle.

According to Figure 7a,b, the HoughLines function performs better in correction than the HoughLinesP function. Further observation of Figure 7b reveals that the overall fluctuation amplitude of the Canny method is the largest, while the Sobel method detects an overall deviation from the true tilt angle value. The Laplacianl and autoCanny methods have some fluctuations in the true tilt angle value, with the Laplacian method having a fluctuation range of 0.8 and the autoCanny method having a fluctuation range of 0.5, indicating that the stability of the autoCanny method is significantly better. However, there are several cases in the autoCanny method where the calculated tilt angle value is 0. This is because, when the camera switches from daytime to nighttime mode, several images with completely black backgrounds appear in the test images. To avoid this situation, a memory strategy can be adopted, which means that, when the current image cannot be recognized, the previously obtained tilt angle is used as the current tilt angle value. Further analysis of Figure 7c shows that, under the influence of the HoughLines function, the MAE and RMSE curves are located at the bottom of the figure. Among these, the MAE values are ordered as follows: Canny > Sobel > Laplacian > autoCanny, whereas the RMSE values are ordered as Canny > Sobel > autoCanny > Laplacian. In terms of MAE, the value for autoCanny is the smallest, indicating that the overall deviation between the tilt angle obtained by autoCanny and the true tilt angle is minimal. However, in terms of RMSE, autoCanny is greater than Laplacian. Therefore, based on the above analysis, this article selects the autoCanny HoughLines as the correction algorithm and applies it to the experiments in this study.

### 4.3. Water-Level Gauge Segmentation Experiment

At present, the YOLO algorithm has spawned multiple series, and there are differences in performance among different series of YOLO algorithms. Due to our use of the YOLO algorithm for tasks such as segmentation and binary classification, we chose mAP: 0.5–0.95 and Val/box_loss as evaluation metrics for the experiment. For the deployment of the model, we chose YOLOv5s and YOLOv5m, which have smaller model sizes and better performance, as baselines and improved their attention mechanisms. We have tried different attention modules, such as C3SE Improved (Bottleneck) C3ECA, C3Ghost, C3SPP, etc. From the results in Figure 6, we can clearly see that the performance of Improved YOLOv5m is significantly better than other models.

From Figure 8a, it can be observed that Improved, C3Ghost, and C3SPP achieved lightweight effects on the YOLOv5m model, while only C3Ghost and C3SPP achieved lightweight effects in the YOLOv5s model.

In Figure 8b, the YOLOv5m model with added attention mechanism shows a positive optimization effect, while the YOLOv5s model shows a negative optimization effect. Among them, the Improved YOLOv5m performs the best on the mAP: 0.5–0.95 index, reaching 0.918, which is 0.016 higher than YOLOv5m.

In Figure 8c, C3SE, Improved, and C3Ghost have had a positive optimization effect on YOLOv5m, while, in terms of YOLOv5s, the optimization effects of Improved, C3Ghost, and C3SPP are relatively small. Similarly, the Improved YOLOv5m performs the best on the Val/box_loss index, with a value less than 0.0028, a decrease of 0.00013 compared to YOLOv5m. This further proves the effectiveness of the Improved YOLOv5m as the main strategy algorithm for water-level monitoring.

Overall, through the improvement of attention mechanisms, the Improved YOLOv5m performs significantly better than other models, with higher mAP and lower Val/box_loss metrics. Therefore, we will use the Improved YOLOv5m as the main strategy algorithm for water-level monitoring.

### 4.4. Water-Level Measurement Experiment

#### 4.4.1. Comparative Experiment

To verify the accuracy and effectiveness of the proposed method, we conducted comparative experiments with traditional image processing methods and previous state-of-the-art methods. As shown in Table 1, the experimental results indicate that traditional image processing methods fail to detect the water level. This may be due to the complex environment of the captured images, which prevents traditional image processing algorithms from detecting the water level in our experiments. This indirectly demonstrates that traditional image processing methods are not suitable for complex environments, exhibiting low robustness. The effect of the binarized image processed by traditional edge detection methods is shown in Figure 9.

**Comparison between different methods**: We compared our proposed method with two previous state-of-the-art methods in Table 1. Our method achieved the best results in Error ≤ 1 cm and Average Error (cm), respectively. The number of instances with Error ≤ 1 cm is 21% higher than the second-best result, and the Average Error (cm) is 0.2 cm lower than the second-best result. This demonstrates that our proposed method offers superior performance.

**Comparison between different water-level conditions**: We conducted detection tests under six different water-level conditions, with the results presented in Table 2. Under Sunny conditions, the images had sufficient lighting and objects were clear, resulting in excellent detection performance. Under Cloudy conditions, although the lighting was slightly dim, the overall image clarity was still sufficient to ensure good detection results. Under Night conditions, with minimal environmental interference and the added optimization from the assisted strategy, we achieved even higher detection accuracy. Under Special conditions, despite anomalies in the detected objects within the images, accurate detection was still possible. However, under Rainy conditions, the presence of significant noise in the images led to a decline in detection performance. Similarly, under Soiling conditions, obstructions caused by dirt slightly reduced detection accuracy. Despite the lower performance under Rainy and Soiling conditions, the average detection error remained below 3 cm, demonstrating that our proposed method maintains a certain level of robustness.

In addition to comparing the proposed algorithm with traditional image processing methods, this article will also conduct ablation experiments.

#### 4.4.2. Ablation Experiment

**Effectiveness of the proposed components**: We incrementally add our contribution in Table 3 and evaluate them in terms of Error distribution and Average Error. Firstly, we individually add Improved YOLOv5m and Assisted strategy. For both components, there is an improvement in the Error distribution and Average Error. Adding all two components gives a clear improvement over the baseline. Particularly, we observe that the upgrades in metrics for the Improved YOLOv5m and the Assisted strategy are roughly equivalent. Finally, adding all components together to our proposed method clearly outperforms all other variants in terms of the Error distribution and Average Error.

**Effectiveness of the proposed components during Day and Night**: We provide the results of the performance improvements for each component during both day and night in Table 4. Firstly, we observe that the Improved YOLOv5m shows improvements in both day and night scenarios, demonstrating the enhanced accuracy of water gauge segmentation. Next, we note that the Assisted strategy significantly improves performance at night, validating its effectiveness in scenarios where the lower part of the water gauge is obscured or the camera fails to capture the image. Overall, the combination of Improved YOLOv5m and the Assisted strategy can effectively enhance the accuracy and stability of the method.

## 5. Conclusions

This article proposes a novel water-level detection method based on computer vision, which combines image processing, deep learning, and machine learning technologies to effectively improve the accuracy, robustness, and applicability of water-level monitoring. We conducted experimental tests on a custom dataset and compared it with traditional image processing-based water-level measurement methods. The experimental results show that the method proposed in this article can automatically and accurately detect water-level values in complex environments, with an average error of less than 3 cm. However, traditional image processing methods cannot detect water-level values in complex environments. This method uses an Improved YOLOv5m to detect water-level gauges in segmented video images and converts the obtained water-level gauge height into water-level values using a linear regression model. Compared with traditional image processing methods, this method has advantages such as high accuracy, strong practicality, and good stability.

Although the height of the water-level gauge segmented by the proposed method is affected by various environmental factors and the image captured by the camera, resulting in significant errors during detection, using context-assisted adjustment can effectively prevent the impact of these factors and improve the stability and accuracy of the algorithm.

In future work, we will further improve the structure of the YOLOv5m model to enhance the accuracy of algorithm segmentation. Alternatively, we could consider using multiple features through linear regression to convert them into water-level values, to prevent the occurrence of significant errors in the predicted water-level values due to deviations in individual features and improve the algorithm’s fault tolerance and stability.

## Figures and Tables

**Figure 1 sensors-24-05235-f001:**
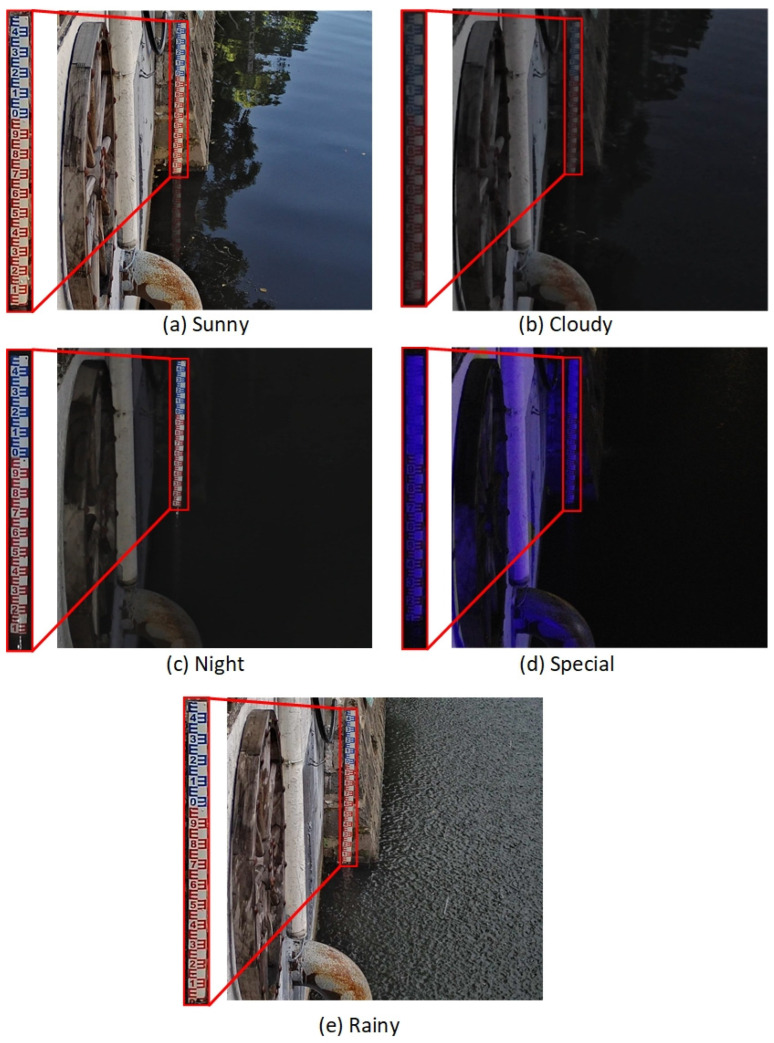
Water-level gauge detection samples in various situations. (**a**) Sunny; (**b**) Cloudy; (**c**) Night; (**d**) Special; (**e**) Rainy.

**Figure 2 sensors-24-05235-f002:**
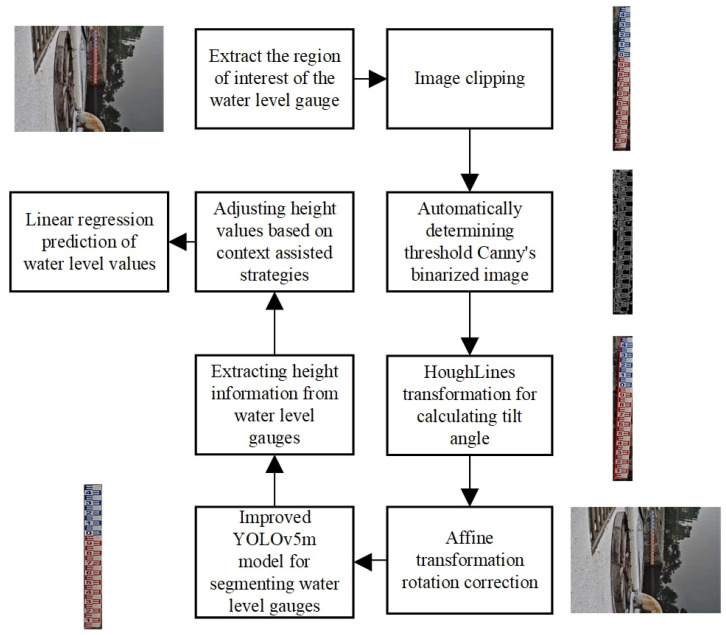
The overall process of water-level measurement.

**Figure 4 sensors-24-05235-f004:**
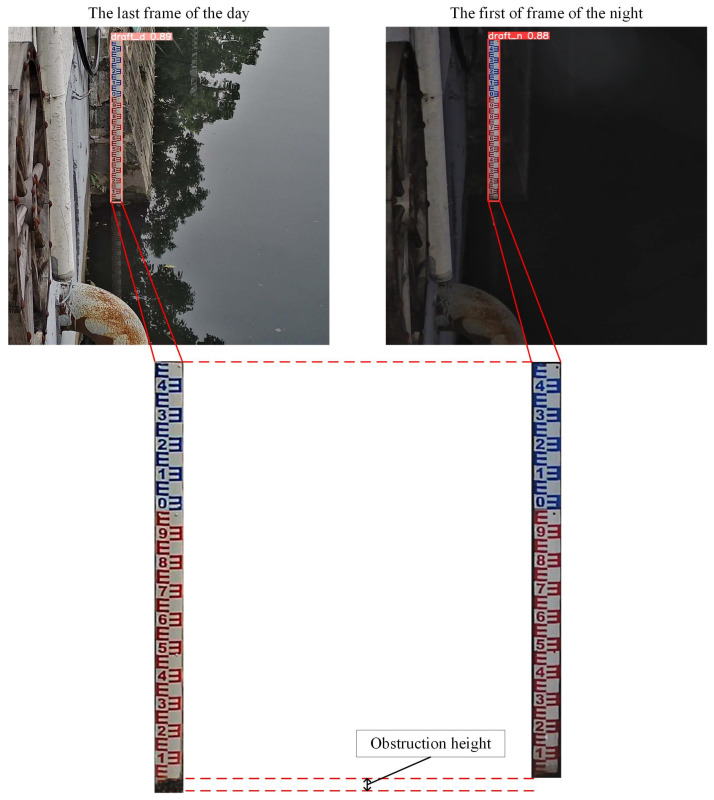
Illustration of water gauge detection in the presence of obstructions: the last frame of the day versus the first frame of the night.

**Figure 6 sensors-24-05235-f006:**
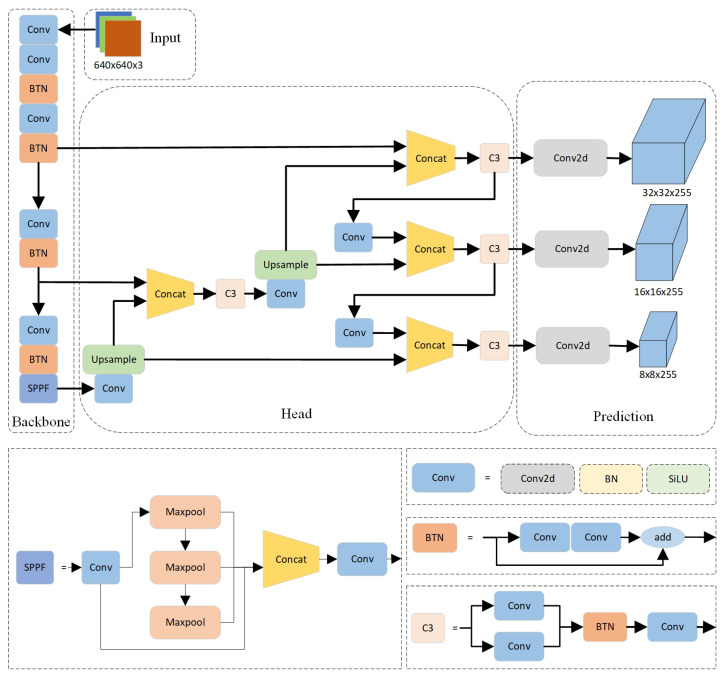
Improved YOLOv5m network framework.

**Figure 7 sensors-24-05235-f007:**
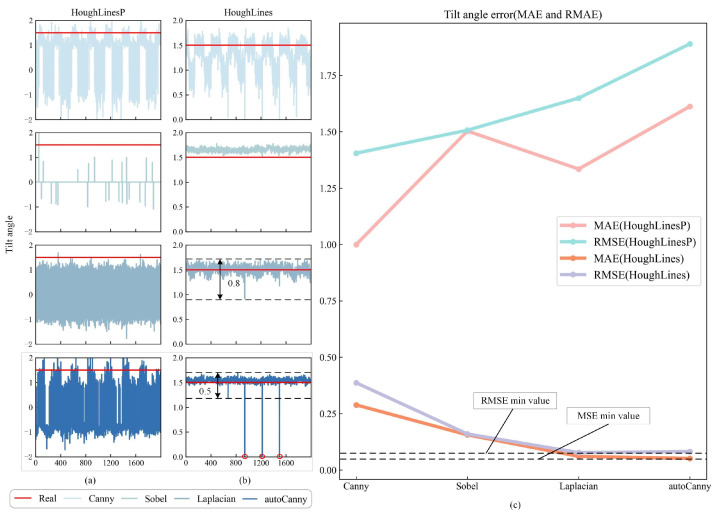
Correction situation of different algorithms. (**a**) The detection performance of the HoughLineP function for identifying tilt angles across four different edge detection operations. (**b**) The detection performance of the HoughLines function for identifying tilt angles across four different edge detection operations. (**c**) The tilt angle error detected by different combination methods.

**Figure 8 sensors-24-05235-f008:**
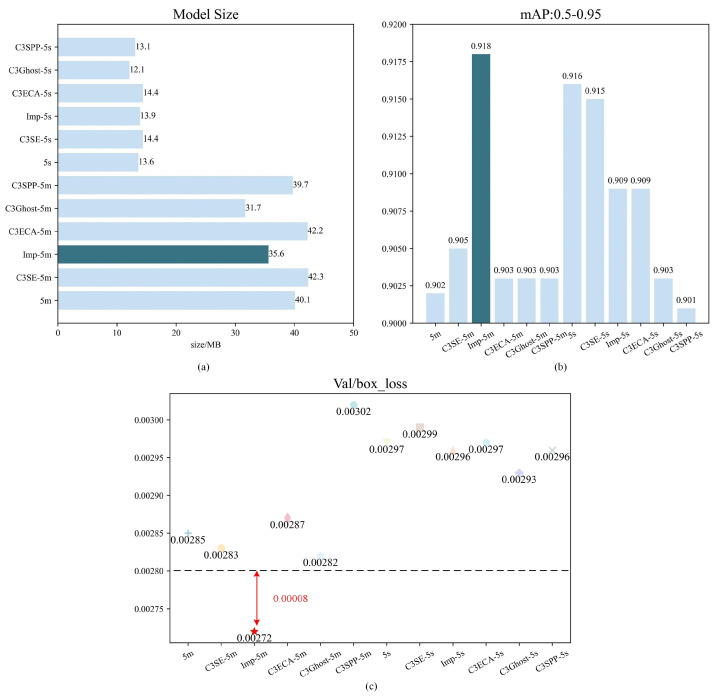
Performance graphs of various YOLO models, comparing their performance in terms of Model size, mAP:0.5–0.95, and Val/box loss. For simplicity, 5m is YOLOv5m, C3SE-5m is C3SE-YOLOv5m, Imp-5m is Improved-YOLOv5m, C3ECA-5m is C3ECA-YOLOv5m, C3Ghost-5m is C3Ghost-YOLOv5m, and C3SPP-5m is C3SPP-YOLOv5m. Similarly, the same goes for YOLOv5s. (**a**) Comparison of model sizes. (**b**) Performance comparison of models based on mAP: 0.5–0.95. (**c**) Performance comparison of models based on Val/box_loss.

**Figure 9 sensors-24-05235-f009:**
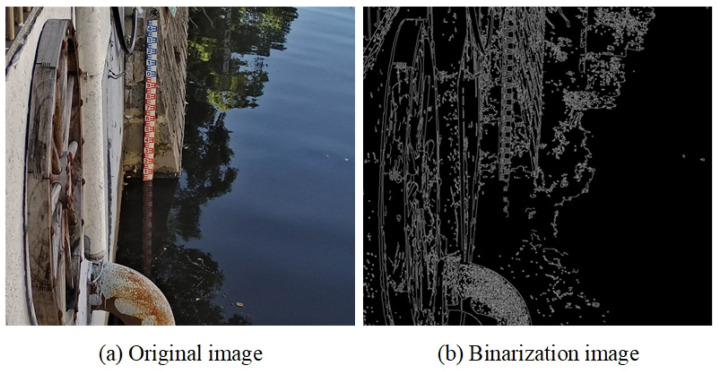
(**a**) Original image; (**b**) binarization image.

**Table 1 sensors-24-05235-t001:** Performance comparison of different methods.

Method	Error ≤ 1 cm	Error ≤ 1 cm	Error ≥ 3 cm	Average Error (cm)
Tradition	~	~	~	~
Qiao [22]	14%	49%	37%	2.6
Chen [24]	25%	56%	19%	1.9
Self	46%	37%	17%	1.7

**Table 2 sensors-24-05235-t002:** Water-level recognition results in different conditions.

Scene	Error ≤ 1 cm	Error ≤ 1 cm	Error ≥ 3 cm	Average Error (cm)
Sunny	58%	42%	0%	0.9
Cloudy	40%	28%	32%	1.9
Rainy	0%	73%	27%	2.2
Night	88%	11%	1%	0.8
Soiling	21%	51%	28%	2.3
Special	100%	0%	0%	0.8

**Table 3 sensors-24-05235-t003:** Ablation study of Improved YOLOv5m and the Assisted strategy.

Method	Improved	Assisted	Error ≤ 1 cm	Error ≤ 3 cm	Error ≥ 3 cm	Average Error (cm)
Self	×	×	12%	43%	45%	2.8
Self	×	√	17%	57%	26%	2.3
Self	√	×	17%	58%	25%	2.2
Self	√	√	46%	37%	17%	1.7

**Table 4 sensors-24-05235-t004:** Ablation study of Improved YOLOv5m and the Assisted strategy during Day and Night.

Method	Scene	Improved	Assisted	Error ≤ 1 cm	Error ≤ 3 cm	Error ≥ 3 cm	Average Error (cm)
Self	Day	×	×	10%	42%	48%	2.8
Self	Night	×	×	8%	44%	48%	2.8
Self	Day	√	×	19%	52%	29%	2.3
Self	Night	√	×	14%	66%	20%	2.3
Self	Day	×	√	9%	45%	46%	2.8
Self	Night	×	√	18%	54%	28%	2.3
Self	Day	√	√	21%	51%	28%	2.3
Self	Night	√	√	88%	11%	1%	0.8

## Data Availability

Data is contained within the article.

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
