# Peer review of "A Complex Environmental Water-Level Detection Method Based on Improved YOLOv5m"

_sensors, 2024, doi:10.3390/s24165235_

Round 1

Reviewer 1 Report

Comments and Suggestions for Authors

This paper proposes a complex environment water level detection method based on an improved YOLOv5m model. By combining computer vision and introducing deep learning target detection technology, it addresses the shortcomings of traditional image-based water level detection methods in complex environments. The paper has good practical application value and provides new technical means for the early warning and monitoring of natural disasters such as floods and droughts. However, there are some issues and deficiencies in the method presented in the article that need to be addressed.

The specific revision suggestions are as follows:

1. The paper only uses affine transformation to correct the rotation of water gauge, without considering the error caused by image perspective distortion due to the tilt camera shooting angle. The paper should provide the range of the shooting angle within which this perspective distortion error can be negligible; otherwise, the practical application scenarios will be greatly limited. The affine transformation formula should also be provided.

2. In line 98 of the paper, the image cropping operation in Figure 2 does not specify the size to which the water gauge region above the water surface should be cropped, considering the varying water levels. Moreover, the operation sequence of cropping first and then rotating is inappropriate, as it may lead to an incomplete water gauge image.

3. In line 160, Hday,∞ is the height of the water gauge in the last frame of the daytime image, Hnight,0 is the height of the water gauge in the first frame of the nighttime image, and Ho is the set threshold. Detailed criteria for determining which frame is the last frame of the daytime image and the first frame of the nighttime image in the video should be provided. Additionally, the setting of this Ho threshold has many uncertain factors, such as lighting, stain, and entanglement of floating debris during floods. The paper should include more detection results under complex scenarios.

4. In line 193, the autoCanny algorithm sets the parameter σ to 0.33. It should be discussed by providing detection images under ideal and non-ideal conditions.

5. The titles of sections 4.3 and 4.4 are the same. Section 4.3 mainly describes the performance comparison of models with different attention mechanisms. However, it does not specify where the attention mechanism is added in the network structure. It is necessary to supplement the network structure diagram with the attention module and conduct experimental analysis of the results, for example, comparing and analyzing some images with large errors to show in which scenarios the optimized model proposed reduces errors compared to other models.

6. Section 4.4 describes the improved model's performance in complex scenarios, but only provides result comparison charts, which do not visually show readers in which complex scenarios the detection accuracy is improved. Continuous water level measurement comparison results under day and night lighting conditions should be supplemented to prove the effectiveness of the proposed model based on the context assistance strategy and attention mechanism. Additionally, some scenarios with large errors should be analyzed.

7. Section 4.4 compares traditional methods with the proposed method. I believe that before using traditional methods, the target area should be determined based on the fixed position of the camera and the water gauge, and then the target area should be binarized, instead of binarizing the whole image, which increases background interference. The paper does not provide specific detection results of traditional methods, only pointing that the results are unsatisfactory.

8. The experimental section does not compare the proposed method with existing image-based water level measurement methods, making it impossible to demonstrate the superiority of the proposed method. The number of literature in the review part is small, it is recommended to supplement the related literature in this field in recent years and do the comparison test, such as “Robust water level measurement method based on computer vision” and “A novel water level recognition method in complex scenes”.

9. In Figure 5a and 5b, the physical meaning of the numbers on the horizontal and vertical coordinates should be explained in the text.

Comments on the Quality of English Language

Minor editing of English language required.

Author Response

 Comments 1: The paper only uses affine transformation to correct the rotation of water gauge, without considering the error caused by image perspective distortion due to the tilt camera shooting angle. The paper should provide the range of the shooting angle within which this perspective distortion error can be negligible; otherwise, the practical application scenarios will be greatly limited. The affine transformation formula should also be provided.

Response 1: Thank you for pointing this out. We agree with this comment. Therefore, we have to explain what change you have made. This revision can be found in lines 108 and 128 of the revised manuscript.

Comments 2: In line 98 of the paper, the image cropping operation in Figure 2 does not specify the size to which the water gauge region above the water surface should be cropped, considering the varying water levels. Moreover, the operation sequence of cropping first and then rotating is inappropriate, as it may lead to an incomplete water gauge image.

Response 2: Thank you for the detailed review. In the operation sequence of cropping first, we use a recognition algorithm to dynamically crop the region of interest (the water gauge section). The purpose of this is to prevent environmental interference with the tilt detection, as the tilt detection process requires binarizing the image. And the rotation operation is not applied to the cropped image but to the entire water gauge image.

Comments 3: In line 160, Hday,∞ is the height of the water gauge in the last frame of the daytime image, Hnight,0 is the height of the water gauge in the first frame of the nighttime image, and Ho is the set threshold. Detailed criteria for determining which frame is the last frame of the daytime image and the first frame of the nighttime image in the video should be provided. Additionally, the setting of this Ho threshold has many uncertain factors, such as lighting, stain, and entanglement of floating debris during floods. The paper should include more detection results under complex scenarios.

Response 3: Thank you for pointing this out. We agree with this comment. Therefore, we have to explain what change you have made. This revision can be found in line 162 of the revised manuscript.

Comments 4: In line 193, the autoCanny algorithm sets the parameter σ to 0.33. It should be discussed by providing detection images under ideal and non-ideal conditions.

Response 4: Thank you for pointing this out. We agree with this comment. Therefore, we have to explain what change you have made. This revision can be found in lines 209 and 218 of the revised manuscript.

Comments 5: The titles of sections 4.3 and 4.4 are the same. Section 4.3 mainly describes the performance comparison of models with different attention mechanisms. However, it does not specify where the attention mechanism is added in the network structure. It is necessary to supplement the network structure diagram with the attention module and conduct experimental analysis of the results, for example, comparing and analyzing some images with large errors to show in which scenarios the optimized model proposed reduces errors compared to other models.

Response 5: Thank you for pointing this out. We agree with this comment. Therefore, we have to explain what change you have made. This revision can be found in lines 284, 348, and 380 of the revised manuscript.

Comments 6: Section 4.4 describes the improved model's performance in complex scenarios, but only provides result comparison charts, which do not visually show readers in which complex scenarios the detection accuracy is improved. Continuous water level measurement comparison results under day and night lighting conditions should be supplemented to prove the effectiveness of the proposed model based on the context assistance strategy and attention mechanism. Additionally, some scenarios with large errors should be analyzed.

Response 6: Thank you for pointing this out. We agree with this comment. Therefore, we have to explain what change you have made. This revision can be found in line 410 of the revised manuscript.

Comments 7: Section 4.4 compares traditional methods with the proposed method. I believe that before using traditional methods, the target area should be determined based on the fixed position of the camera and the water gauge, and then the target area should be binarized, instead of binarizing the whole image, which increases background interference. The paper does not provide specific detection results of traditional methods, only pointing that the results are unsatisfactory.

Response 7: Thank you for pointing this out. We agree with this comment. Therefore, we have to explain what change you have made. This revision can be found in line 382 of the revised manuscript.

Comments 8: The experimental section does not compare the proposed method with existing image-based water level measurement methods, making it impossible to demonstrate the superiority of the proposed method. The number of literature in the review part is small, it is recommended to supplement the related literature in this field in recent years and do the comparison test, such as “Robust water level measurement method based on computer vision” and “A novel water level recognition method in complex scenes”.

Response 8: Thank you for pointing this out. We agree with this comment. Therefore, we have to explain what change you have made. This revision can be found in lines 391 and 398 of the revised manuscript.

Comments 9: In Figure 5a and 5b, the physical meaning of the numbers on the horizontal and vertical coordinates should be explained in the text.

Response 9: Thank you for pointing this out. We agree with this comment. Therefore, we have to explain what change you have made. This revision can be found in line 323 of the revised manuscript.

4. Response to Comments on the Quality of English Language

Point 1: Minor editing of English language required.

Response 1: We have corrected the language and spelling errors.

Reviewer 2 Report

Comments and Suggestions for Authors

The article describes a method for determining the water level, but, unfortunately, it does not describe in which water basins this method is applicable. This is extremely important to take into account all natural and non-natural factors that affect the accuracy of measurements. Remarks: 1. The article practically does not describe modern and non-modern methods for determining the water level in the sea, bay, etc. Only links to the works that the authors need to develop their field are provided. 2. The "recessed" ruler as an integral part of the sensor is an antediluvian tool. There are various methods, also based on various video cameras and computer vision, which are not based on a "ruler" as a scale. 3. Why did the authors use linear regression to recover "unprocessed" or lost data? This is an extremely rough approximation. There are various non-linear natural processes that affect measurements. It is enough to list the following: tides, wind surge, seiches, variations in atmospheric pressure, etc. As the instrumental experience shows, the use of a fifth or sixth degree polynomial is most effective for restoring missing data. Resume: 1. The literature review is very poor. 2. To demonstrate in the literature review other methods with better qualities, based also on video images and computer vision. How is your method better? His accuracy is worse. 3. To prove the applicability of linear regression in comparison with polynomials of a different degree (up to the sixth).

Author Response

Comments 1: The literature review is very poor.

Response 1: Thank you for pointing this out. We agree with this comment. Therefore, we have to explain what change you have made. This revision can be found in line 60 of the revised manuscript.

Comments 2: To demonstrate in the literature review other methods with better qualities, based also on video images and computer vision. How is your method better? His accuracy is worse.

Response 2: Thank you for pointing this out. We agree with this comment. Therefore, we have to explain what change you have made. This revision can be found in line 382 of the revised manuscript.

Comments 3: Why did the authors use linear regression to recover "unprocessed" or lost data? To prove the applicability of linear regression in comparison with polynomials of a different degree (up to the sixth).

Response 3: Thank you for the detailed review. We use linear regression not to recover “unprocessed” or lost data. We use linear regression to convert the pixel height information of the water gauge in the image into actual water level values.

Round 2

Reviewer 1 Report

Comments and Suggestions for Authors

The authors have made a sincere and substantial effort towards increasing the quality of the manuscript. They have satisfactorily addressed most of my concerns. From my point of view, the paper could be considered ready for accept after minor rivision of following two issues.

1. In line 111, "the perspective distortion in images captured within the central area of the camera, approximately 50-60 degrees can be ignored". How do you get to this conclusion ?Error analysis  is recommended.

2. Considering the performance of deep learning methods depends on training samples. It is suggested to supplement the water level distribution of dataset in  in Section 2.1, and discuss the measurement results under different water level conditions in the experiment.

Author Response

Comments 1: In line 111, "the perspective distortion in images captured within the central area of the camera, approximately 50-60 degrees can be ignored". How do you get to this conclusion ?Error analysis  is recommended

Response 1: We use Hikvision cameras equipped with standard wide-angle lenses. In the central area of the camera (approximately 50-60 degrees), perspective distortion in the captured images can be ignored due to the camera's performance.

Comments 2: Considering the performance of deep learning methods depends on training samples. It is suggested to supplement the water level distribution of dataset in  in Section 2.1, and discuss the measurement results under different water level conditions in the experiment.

Response 2: Thank you for pointing this out. We agree with this comment. Therefore, we have to explain what change you have made. This revision can be found in lines 85 and 402 of the revised manuscript.

Reviewer 2 Report

Comments and Suggestions for Authors

Thanks for the corrections. I think the article can be criticized.

Author Response

Thank you very much for your comments.